# Blockchain Censorship

## ABSTRACT

Permissionless blockchains promise resilience against censorship by a single entity. This suggests that deterministic rules, not third-party actors, are responsible for deciding whether a transaction is appended to the blockchain. In 2022, the U.S. Office of Foreign Assets Control (OFAC) sanctioned a Bitcoin mixer and an Ethereum application, challenging the neutrality of permissionless blockchains.

In this paper, we formalize, quantify, and analyze the security impact of blockchain censorship. We start by defining censorship, followed by a quantitative assessment of current censorship practices. We find that 46% of Ethereum blocks were made by censoring actors complying with OFAC sanctions, indicating the significant impact of OFAC sanctions on the neutrality of public blockchains.

We uncover that censorship impacts not only neutrality but also security. After Ethereum's shift to Proof-of-Stake (PoS), censored transactions faced an average delay of 85%, compromising their security and strengthening sandwich adversaries. Finally, we prove a fundamental limitation of PoS and Proof-of-Work (PoW) protocols against censorship resilience.

### ACM Reference Format:
Anonymous Author(s). 2023. Blockchain Censorship. In *Proceedings of ACM Conference (Conference'17)*. ACM, New York, NY, USA, 12 pages. https://doi.org/10.1145/nnnnnnn.nnnnnnn

## 1 INTRODUCTION

Permissionless blockchains enable participants to transact with each other without the need for a trusted intermediary. In theory, pseudonymous users can use permissionless blockchains without anyone being capable of censoring or seizing control of the network.

Related works already study the use of blockchain applications for money laundering and other illicit purposes [55, 59]. These malpractices attracted the attention of governments. The U.S. Office of Foreign Assets Control (OFAC) included blockchain addresses in its Specially Designated Nationals and Blocked Persons (SDN) list. Those subject to the U.S. jurisdiction are prohibited from interacting with persons and property on the SDN list. OFAC sanctioned the cryptocurrency service provider Blender.io on May 5, 2022, for using its privacy-enhancing technology to facilitate criminal money laundering. This was followed by the sanctioning of Tornado Cash (TC) on August 8, 2022, for the same reason. Blender.io is a centralized service for hiding Bitcoin (BTC) money flows, requiring users to trust those managing the service. In contrast, TC is an autonomous and immutable smart contract application on Ethereum [51]. The imposed OFAC sanctions against smart contract addresses are unprecedented and resulted in cryptocurrency providers restricting users from using their services [75].

**This paper.** We provide a holistic overview of blockchain censorship (§ 4). We focus on censorship on the consensus layer, as validator nodes are responsible for including transactions in a block, and censorship on the application layer, as smart contracts can prevent the successful execution of transactions in a block (§ 4.3).

We analyze the impact of OFAC sanctions on Ethereum before (§ 5.2) and after (§ 5.3) the transition to Proof-of-Stake (PoS) ("the merge"). We show that interactions with TC's smart contracts declined by 84.3% within two months following the sanctions. We demonstrate that Ethermine, commanding 22% of Ethereum's Proof-of-Work (PoW) hash rate, excluded TC interactions from their blocks, leading to a daily reduction of 200 blocks ($\sim$ 33.4%) containing TC transactions. For post-merge Ethereum, we find that over two months, at least 46% of the total blocks were made by actors engaged in transaction censorship due to OFAC sanctions. At the application layer, we observed a spike in blocked users by 84.99% in August 2022, the month of introducing the OFAC sanctions. On Bitcoin, we find that OFAC sanctions prevented the Bitcoin mixer Blender.io from continuing to provide its centralized services (§ A).

We also study the implications of censorship on blockchain security (§ 6). We find that censorship delays the inclusion of *both* censored and non-censored transactions by increasing their time in the memory pool (i.e., mempool) (§ 6.1). Finally, we prove that if > 50% of validator nodes directly censor transactions, a PoS blockchain cannot achieve censorship resilience (§ B). To the best of our knowledge, we are the first to provide an empirical overview of applied censorship measures (§ 5) and associated security implications (§ 6). Thereby, our contributions are threefold:

- We define blockchain censorship across system layers and temporal features, quantitatively analyzing censorship by block builders, proposers, relayers and smart contracts.
- We provide quantitative evidence of the historical transaction confirmation latency on Ethereum. We find that Ethereum's move to PoS and Proposer-Builder Separation (PBS) has been delayed, including both censored and non-censored transactions. E.g., the average inclusion delay for TC transactions increased from 15.8 ± 22.8 seconds in Aug. 2022 to 29.3 ± 23.9 seconds in Nov. 2022. Increased confirmation latencies exacerbate sandwich attack risks.
- We prove that no PoS (PoW) protocol can achieve *censorship-resilience* if the censoring validators (miners) make up more than 50% of the validator committee (hashing power).

## 2 BACKGROUND

### 2.1 Permissionless Blockchains

Permissionless blockchains build upon the premise of relying on a deterministic set of rules instead of trusted parties to determine the validity of a transaction. Bitcoin [43] is the first permissionless blockchain that enables any entity to create transactions and broadcast them to miners which eventually include them in a block appended to the blockchain. For the most part, Bitcoin transactions represent monetary flows between peers, though it is also possible to write arbitrary data onto the Bitcoin blockchain. Ethereum [74] goes further than Bitcoin by allowing the deployment of arbitrary

*Conference'17, July 2017, Washington, DC, USA*
2023. ACM ISBN 978-x-xxxx-xxxx-x/YY/MM...$15.00
https://doi.org/10.1145/nnnnnnn.nnnnnnn

code, commonly referred to as *smart contracts*, to the blockchain, which is then executed in a decentralized manner. Smart contracts gave birth to a thriving ecosystem of financial applications, Decentralized Finance (DeFi). Competitive trading on DeFi emerged along with novel attacks [84], such as sandwich attacks [83], more generally, front- and back-running [53, 54] exploiting transaction ordering for a financial gain. Bitcoin relies on PoW [4], while Ethereum switched to PoS in September 2022 during "the merge" [15].

**Proposer/Builder Separation.** Shortly after transitioning to PoS, PBS was introduced to Ethereum. PBS separates the functions of creating new blocks and appending blocks to the blockchain. This is a direct response to problems associated with Miner/ Maximum Extractable Value (MEV) [12, 54], and should supposedly enhance Ethereum's censorship-resistance [16]. MEV extraction can negatively affect user experience [71], and more importantly, the underlying incentive structure of the blockchain, thereby harming blockchain security [54, 79, 81]. In PBS, the role of a "validator" ("miner" in PoW) is divided between separate entities, namely "block builders" and "block proposers" (i.e. the validators themselves). In addition, "relays" were introduced to intermediate and establish the required trust between block builders and proposers. Currently, it is optional for Ethereum validators to participate in PBS, and they can do so by using software called MEV-Boost. Validators are still free not to use MEV-Boost and to build blocks independently.

**Privacy-Enhancing Technologies.** In blockchain systems such as Bitcoin and Ethereum, asset transfers are transparently traceable [1, 70]. For example, "mixing services" enable obfuscation of asset flows by creating shared transactions with other users or routing assets through shared addresses [24]. For Bitcoin, Blender.io is an example of an application that attempts to enhance privacy by allowing users to deposit their assets into a shared account together with other users and later withdraw them to a newly created, pseudonymous account. Unlike Blender.io, CoinJoin wallets do not require users to trust a service operator [38].

On Ethereum, TC represents the most prominent example of a privacy-enhancing application [51]. TC allows users to deposit assets into a shared account and later withdraw the assets anonymously to a newly generated address, thereby preventing observers from tracing asset flows [32]. This is achieved by relying on Zero-Knowledge Succinct Non-Interactive Argument of Knowledge (zk-SNARKS) [6]. TC offers different "pools" in which users can deposit assets of a fixed denomination, such as a 0.1, 1, 10, or 100 ETH. Users who deposit funds into a given denomination's contract can later withdraw the same amount from the respective pool without revealing their deposit address. The fixed denomination aims to obscure the link between deposits and withdrawals for observers.

Governments believe entities like the North Korean Lazarus Group used the aforementioned privacy technologies for money laundering and evading sanctions [49].

## 2.2 Rationales for Censorship

While permissionless blockchains aim to resist censorship, real-world pressures can challenge this objective. Certain actors within the blockchain can obstruct user transactions or even prevent block finalization, driven by various motivations. Some reasons for obstruction are *external*, such as government or legal pressures. Others are *internal*, driven by ethical considerations or economic gain.

Endogenous and exogenous reasons are, in practice, intertwined. For example, assume that in some jurisdictions, it is unclear whether the law requires node operators to obstruct blockchain transactions involving addresses linked to a specific criminal organization. In this situation, a node operator may decide to obstruct for many reasons, including (1) lowering their risk of legal liability, (2) a genuine desire not to facilitate criminal activity, and (3) economic motives (e.g., appearing as a responsible business operator to investors). Only the first reason is exogenous, and even then, whether any operator will act on that reason will depend on their risk tolerance (an endogenous factor), especially given that the law is unclear.

**Legal and Political Rationales.** Blockchain transactions are sometimes used for purposes that are criminalized in most jurisdictions, like hacking, theft, or payments facilitating crimes (e.g., for Child Sexual Abuse Materials [11, 45], drugs and dark web markets [9]). Also, blockchain transactions are used for purposes prohibited for national security or humanitarian reasons, where there is less convergence across various jurisdictions. The key case of the latter is violations of economic sanctions laws. Targeting senders or addressees of such transactions may be challenging for law enforcement. Hence, from the perspective of preventing and deterring legally undesirable behavior, imposing legal obligations to censor transactions may seem attractive.

**The U.S. Economic Sanctions.** Under U.S. sanctions law, it is prohibited to engage in transactions with sanctioned entities, their property, and their interests in property [69]. It is also prohibited to make "any contribution or provision of funds, goods, or services by, to, or for the benefit of" to sanctioned entities. [52] The U.S. OFAC maintains the Specially Designated Nationals and Blocked Persons (SDN) list, including blockchain addresses of sanctioned persons and organizations since 2018. [69] We refer to addresses included in the SDN list as "sanctioned addresses." On a technical note, Ethereum addresses (accounts) tend to prove more persistent than Bitcoin addresses (UTXO). [70]

We focus on two sanctions designations made by OFAC in 2022: Blender.io in May and Tornado Cash in August (re-designated in November) [66–68]. In both cases, blockchain addresses were added to the SDN list. Notably, in the TC case, some SDN-listed addresses refer to smart contracts without administrative functionality. This was the first time smart contract addresses were added to the SDN list. OFAC later clarified that the open-source code of TC smart contracts is not in itself sanctioned, only its instances deployed by the Tornado Cash organisation. [65]

The TC sanctions motivated blockchain node operators to censor transactions involving addresses on the SDN list (cf. Section 5). However, it is subject to debate whether censorship by blockchain validators, block builders, or relays is required by law [56]. Moreover, *if* those network participants are legally required to censor, it may be insufficient only to censor addresses on the SDN list without attempting to censor unlisted addresses used by sanctioned entities, as OFAC clarified that the SDN list is not exhaustive in this respect [69]. When we use the term "OFAC-compliance," we do so informally, referring to the likely rationale of the actor in question while allowing for the possibility that their actions are not legally required or insufficient for compliance with U.S. sanctions law.

## 3 RELATED WORK

**Defining Censorship.** Related contexts in which "censorship" has been defined, at least indirectly (negatively), include works on censorship-resistance (circumvention) in information systems in general [13, 30, 33, 64], and works on censorship-resistance of blockchains in particular [10, 28, 58, 61, 77, 80]. In previous works "censorship" has been understood as: *(a)* not including blocks with transactions to or from targeted entities [10, 58]; *(b)* publicly announcing an intent to exclude future transactions of targeted entities, e.g., by *feather forking* [80]; *(c)* refusing to attest to a chain that contains transactions from or to a targeted entity [58, 61].

The first kind of censorship may apply to block contents or the entity that mined or proposed the block [61]. Censorship of the second kind may also apply to the block content or the identity of the respective user. The third point is specific to PoS-based blockchains [48]. We focus on selective censorship within a network, instead of censorship of entire networks [50].

**Censorship Attacks.** The literature explores multiple attack vectors relevant to censorship, ranging from Denial of Service (DoS) [5], eclipse [23, 25], routing [2], to prefix hijacking [63]. Focusing on censorship on the consensus layer, Miller [40] introduced the *feather forking* attack, where attackers with a minority of the hash-rate in a PoW blockchain can censor transactions, which was later expanded upon by McCorry *et al.* [39], who propose methods to censor confirmed and unconfirmed transactions. Regarding the possibility of censorship at the network layer, Loe *et al.* [35] show that two methods to join a cryptocurrency network, DNS seeding and IP hard-coding, are vulnerable to censorship.

**Censorship Examples.** As part of an attack or due to legal obligations, an entity may be ignored or even blocked by others. Remote Procedure Call (RPC) endpoints can prevent users from broadcasting their transactions, e.g., in March 2022 the Ethereum RPC endpoint Infura censored OFAC-sanctioned entities [46]. In the front-end, wallet applications have been implicated in censoring transactions [46], and similarly, the web applications of DeFi projects have refused to engage with users who received funds from TC [75]. At the consensus layer, it was reported that a mining pool suppressed the inclusion of Initial Coin Offering transactions [14]. A temporal delay in the execution of a transaction may entail significant financial implications for the censored entity [83].

**Preventing Censorship.** Zhang *et al.* [80] propose a multi-metric evaluation framework for quantifying the attack resistance of PoW-based blockchains, including against feather-forking attacks. Kostiainen *et al.* [31] develop a censorship-resistant and confidential payment channel that can be deployed to EVM-compatible blockchains. Le and Gervais [32] construct a reward-enabled censorship-resilient mixer. Lotem *et al.* [36] present a mechanism for on-chain congestion detection which can partially defend against censorship attacks. Karakostas *et al.* [28] present a method to assess blockchain decentralization, asserting that centralization can undermine censorship-resistance in permissionless protocols.

We build upon prior research to provide a quantitative overview of censorship on public and permissionless blockchains. To the best of our knowledge, we are the first to quantify censorship by different ecosystem actors and discuss its security ramifications.

## 4 OVERVIEW OF CENSORSHIP

We proceed to outline our system model and provide a definition of censorship on permissionless blockchains.

### 4.1 System Model

We extend the system model of Zhou *et. al* [84]:

**Network Layer.** In a blockchain, validators form a P2P network by following a set of rules which determine the communication interface, peer discovery as well as procedures for joining and exiting the network. Messages are transmitted between network participants via e.g., gossip or dedicated communication channels. A user may include its message (or "transaction") in the blockchain by joining the P2P network through a self-operated node or by relying on intermediary services (i.e., RPC providers).

**Consensus Layer.** On the consensus layer, a fault-tolerant consensus algorithm ensures that validators in the P2P network are in agreement on a shared state. In a blockchain, a newly proposed block is appended by the validator which is elected through a leader election protocol (e.g., PoW). A block consists of transactions, where the node appending the block to the blockchain decides on the order of included transactions. Nodes are incentivized through a *block reward*, paid for validating a block, and a *transaction fee*, which is paid by the client. Each included transaction advances the shared network state, which is replicated by each validator.

**Application Layer.** *Decentralized applications* (i.e., smart contracts), are smart contracts that maintain a state. A smart contract is defined by a set of functions that cause state transitions and can be invoked through a transaction. A smart contract can interact with other contracts through *internal calls*. While there is no limit on the number of contracts a contract can interact with, blockchains specify an upper limit on the number of instructions a transaction can execute (e.g., the *gas limit* in Ethereum).

**Auxiliary Services.** Auxiliary services are e.g., browser-based cryptocurrency wallets, user interfaces of decentralized applications, and off-chain oracles.

### 4.2 Notation & Terminology

In this work, we assume a single blockchain $\mathcal{L}$ consisting of blocks $B_i$, where $i$ corresponds to a *block identifier*, with $h$ corresponding to the block height. We say that $B_i \in \mathcal{L}$, if a block $B_i$ is included in the blockchain $\mathcal{L}$. The blockchain $\mathcal{L}$ is maintained by a set of $n$ validators, which agree upon the current state of $\mathcal{L}$ through a State Machine Replication (SMR) protocol $\Pi$. The protocol $\Pi$ receives as an input a set of transactions $\mathbf{tx}$, and outputs the ordered ledger of transactions $\mathcal{L}$. Let $\sigma$ be a security parameter that determines the finality of $\mathcal{L}$. Then, we denote $T_\Delta$, a polynomial function in $\sigma$, as the *finality delay*. We define *transaction inclusion* as follows:

*Definition 1 (Transaction Inclusion).* A transaction $tx$ received by a validator at time $t$ is *included* in $\mathcal{L}$ by the SMR protocol $\Pi$, if $tx \in B_i \mid B_i \in \mathcal{L}$ at time $t' > t + T_\Delta$.

Further, we denote the address of an account maintained through $\mathcal{L}$ as $a_i$. We intentionally do not differentiate between externally owned accounts and smart contracts, as this abstraction is irrelevant concerning censorship. When preventing censorship, we differentiate between censorship *resistance* and censorship *resilience*.

Censorship resistance describes a technology that prevents protocol participants from censoring (e.g., confidential "to" addresses). In contrast, censorship resilience describes that censorship is possible for an individual, but the respective system is resilient against it.

### 4.3 Definition of Censorship

Censorship is a broad term that may apply in any system layer as introduced in Section 4.1. To clarify the notion in a blockchain context, we set out to synthesize existing notations in formal definitions.
**Consensus Layer Censorship.** Censorship on the consensus layer may either be enforced *directly* or *indirectly*. For example, a validator may enforce direct censorship by refusing to broadcast a received transaction, sign an attestation, or include a transaction in a block (cf. § 5). Alternatively, an external entity may indirectly enforce censorship by preventing the timely transmission of messages or occupying validator nodes through targeted DoS attacks. Hence, censorship on the consensus layer can also indirectly originate from the network or application layers. Therefore, we focus our definition of censorship on the consensus layer on the *intent* of a protocol participant to obstruct the inclusion of a transaction.

*Definition 2 (Strict Censorship).* A transaction is censored if a protocol participant intentionally obstructs the inclusion of a transaction, such that $tx \notin B_i \mid B_i \in \mathcal{L}$.

Furthermore, we identify a subtle variant of censorship, where transactions are included with a delay.

*Definition 3 (Weak Censorship).* A transaction is censored if an actor intentionally obstructs the inclusion of a transaction in the next possible block, such that a transaction $tx$, received at block height $h$, does not get included in a block $B_i$ at block height $h' = h+1$, thus $tx \in B_i \mid B_i \in \mathcal{L}$, yet $h_{B_i} < h'_{B_i}$.

Definition 2 and Definition 3 follow related works in distinguishing *censorship* from *ordering* of transactions [3, 7, 8, 26, 54]. As transactions are only ordered once decided that they are included in a block, we do not treat it as "censorship" when block builders order ("re-order") transactions differently than expected by the senders of these transactions (e.g., in Ethereum, there may be an expectation of ordering only according to the fees paid by each transaction, which is the default behaviour [19]). We note, however, that intentional re-ordering may result in the transaction failing or in a lower economic gain for the user (e.g., due to front-running [54]).

**Application Layer Censorship.** As previously defined, "strict censorship" cannot be enforced directly by the application layer, i.e., smart contracts, because they cannot directly affect the inclusion or finalization of blocks and transactions at the consensus layer. However, a smart contract can indirectly influence blockchain censorship by incentivizing the inclusion or exclusion of transactions and blocks or incentivizing retroactive forking of a blockchain [39, 44, 73]. Besides indirectly influencing the consensus layer, smart contracts can enforce direct censorship by preventing the successful execution of transactions included in a block. We define *smart contract censorship* as follows.

*Definition 4 (Smart Contract Censorship).* A transaction $tx$ is censored by a smart contract, if $tx \in B_i$, where $B_i \in \mathcal{L}$ is blocked by the state $st_i$, s.t. further state transitions $st_i \rightarrow st_{i+1}$ are blocked by the respective contract.

An example of smart contract censorship is a block list, which prevents an account with address $a_i$ from successfully interacting with the block listing smart contract (cf. Section 5).

## 5 CENSORSHIP QUANTIFICATION

In the following, we provide an empirical quantification of censorship on Ethereum and Bitcoin. We distinguish pre- and post-merge Ethereum as the consensus mechanism impacts censorship.

### 5.1 Data Collection

We collect data about the OFAC-sanctioned applications TC and Blender.io starting from the 1st of January 2021 00:00:00 UTC until the 15th of November 2022 23:59:59 UTC.

**Blender.io Data.** For data on Blender.io, we set up a local Bitcoin node and parse the raw data files. We filter for transactions from and to the sanctioned addresses of Blender.io using the addresses listed in OFAC's SDN list.

**TC Data.** For collecting Ethereum application layer data, we connect to an RPC provider Infura and leverage the Etherscan API. This includes event logs broadcast by sanctioned TC contracts. The event logs indicate that a user has either deposited or withdrawn funds from a TC contract. We include all existing TC pool-contracts in all denominations (cf. Table 2). Notably, we did not include all sanctioned addresses, instead, we focus on deposits and withdrawals to the known TC pool contracts. This means that, e.g., the TC Gitcoin grant contract, contracts deployed on Layer-2 solutions such as Polygon or Arbitrum, or contracts only existing on an Ethereum testnet have been ignored. However, we capture most of the traffic from sanctioned entities because most users interact with the ETH-denominated contracts deployed to the Ethereum mainnet. In total, our data set has 273,403 entries, each representing either a TC deposit or withdrawal, included in 236,868 distinct blocks.

**Ethereum Ecosystem Data.** We collect data on the different ecosystem participants, such as miners, block proposers, block relayers, and block builders. For information on external actors such as block builders and relay operators, we use the Relay Data API. The ProposerPayloadsDelivered API endpoint enables us to retrieve information on the parties involved in PBS. In particular, we are interested in the blocks that block builders deliver to proposers. We connect to every existing relay provider by November 2022 including Flashbots, BloXroute, Blocknative, Manifold, Eden, and Relayooor. Summarizing, our final data set contains 443,831 blocks, which includes every block since PBS was launched until the 15th of November 2022 23:59:59 UTC.

**OFAC SDN List.** At the time of writing, OFAC's SDN list includes 132 Ethereum addresses. 90 (68%) of the sanctioned addresses belong to the privacy tool TC.

In the following sections, we start by identifying the effects of the sanctions on the TC contracts by assessing their immediate impact on user engagement. Second, we focus on the effects of the sanctions on the individual validators. Third, we assess the impact of the sanctions on the distinct participants of the ecosystem. Thus, we distinguish between block builders, proposers, and relayers.

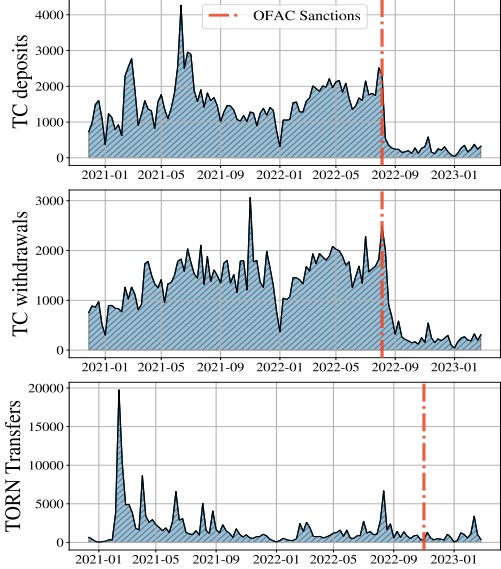

**Figure 1: TC deposits and withdrawals over time, accounting for all deployed TC contracts in every denomination from the 1st of October 2021 to November 15th, 2022.**

## 5.2 Pre-PBS Consensus-Layer Censorship

Figure 1 shows the number of interactions with TC contracts over time through the number of weekly deposits and withdrawals. While the weekly deposits and withdrawals reached over 2,000 before the sanctions, TC's activity afterward reduced by ten-fold to about 200 deposits and withdrawals per week. As of the enactment of the sanctions, we observe a decline in interactions with TC contracts. For October 2022, a total of 1,630 interactions were observed, compared to 16,347 interactions in July 2022. However, notably, the number of interactions has never dropped to zero.

A decline in activity weakens TC's anonymity set, as user privacy hinges on collective participation [72]. In TC, more users amplify individual privacy due to network effects. Reduced anonymity sets heighten the risk of user deanonymization via side channels.

A reasonable explanation for the decrease in interactions with TC's contracts is that due to the sanctions, the TC website was promptly taken off-line [42]. Consequently, users could only interact with the respective contracts without using any interface, which may not have been feasible for most users. In addition, the open-source Github repository that hosts the TC code was temporarily taken offline, preventing users from redeploying the front end. Circle, the company issuing the USDC stablecoin, froze all USDC tokens inside the TC contracts. As a result, the owners of those assets can no longer move their funds.

**How Miners React to Sanctions.** Shifting the focus to the largest miners, that eventually decide upon the inclusion of TC transactions into their blocks, we visualize the number of uncensored blocks over time from July 1st, 2021 to September 15th, 2022 in Figure 2. We observe a decrease in uncensored blocks for the 10 largest miners, which is partly an expected consequence of the

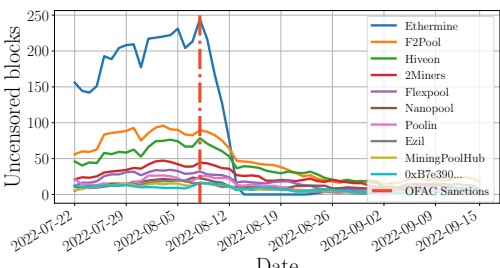

**Figure 2: Blocks containing TC transactions by the top 10 miners of uncensored blocks from July 22, 2022 until the transition to PoS on September 15, 2022 (5 day average).**

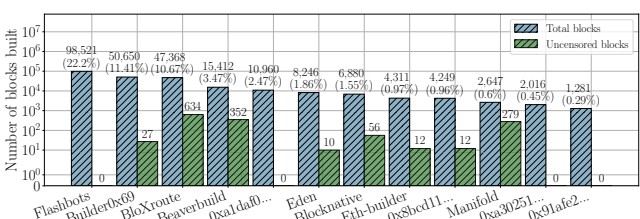

**Figure 3: Block builders on the Ethereum blockchain since the activation of PBS at block height 15,537,940 (Sep-15-2022 08:33:47 AM) until block height 15,978,869 (Nov-15-2022 12:59:59 UTC). Uncensored blocks represent blocks containing interactions with TC contracts.**

overall decrease in TC transactions. Nevertheless, Figure 2 indicates that the decline has been more pronounced for Ethermine compared to other miners. Before the sanctions, we observe, on average 608 (8.5%) blocks containing uncensored transactions per day. Before the sanctions, on average, 203 uncensored blocks per day were built by Ethermine, representing ∼ 33.4% of the total number of uncensored blocks per day. After the sanctions, the number of uncensored blocks built by Ethermine decreased to ∼ 21 blocks per day, which yields a reduction of almost 90%. For the remaining miners, we observe a decrease of uncensored blocks between 50% and 65%, a significantly smaller decline, while no miner altogether ceased, including TC transactions.

## 5.3 Post-PBS Consensus-Layer Censorship

On September 15th, 2022, 38 days after the TC sanctions, Ethereum transitioned to PoS and partially adopted PBS, adding new intermediaries to the ecosystem. Block builders, proposers, and relayers have distinct responsibilities and methods to censor the Ethereum blockchain. By November 15, 2022, with growing PBS adoption, third-party block builders constructed 58% of all blocks. We subsequently segment the next section by participant and analyze censorship for each of them separately.

**Block Builder Censorship.** External block builders take bundles of transactions, construct blocks and pass them to block proposers. We display the ten largest block builders along with their total number of blocks proposed in Figure 3. We add the total number of uncensored blocks to reveal potential censorship practices.

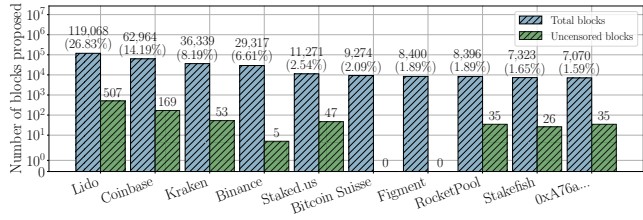

**Figure 4: Ethereum block proposers from PBS activation at block 15,537,940 (Sep-15-2022 08:33:47 AM) to block 15,978,869 (Nov-15-2022 12:59:59 UTC). Uncensored blocks contain transactions interacting with TC contracts.**

Figure 3 shows that Flashbots' block builders are the most successful, as measured by the number of blocks they created. Flashbots' builders are responsible for ~ 22.2% of all blocks created between the PoS transition and the $15^{th}$ of November 2022. This culminates to 97,324 blocks in that timeframe. The builders of Builder0x69 are the second most successful with a total of 50,650 (~ 11.41%) blocks, followed by BloXroute, accounting for 47,368 (~ 10.67%) blocks, and Beaverbuild with 15,412 (~ 3.47%) blocks.

Our results suggest that the four largest block builders of the Ethereum network engage in censoring by not including deposits *to* and withdrawals *from* the TC contracts. The same applies to one of the BloXroute builders, accounting for 2.2% of the total number of blocks built, as well as the anonymous builder with the public key 0xa1daf0..., responsible for 2.5% of the total number of blocks.

Among the most successful builders in Figure 3, only three include TC deposits and withdrawals in their blocks. Two belong to BloXroute and one belongs to Beaverbuild.

**Block Proposer Censorship.** We visualize the most successful block proposers in Figure 4. The staking pool Lido is the most successful group of block proposers between the launch of PBS and the $15^{th}$ of November 2022, proposing 119,068 (~ 26.83%) valid blocks. The $2^{nd}$, $3^{rd}$ and $4^{th}$ most successful block proposers are the exchanges Coinbase, Kraken, and Binance with a total of 62,964, 36,339 and 29,317 blocks respectively. Additionally, we identify staking pools such as Staked.us, Figment, Rocketpool, and Stakefish as among the most successful block proposers.

Focusing on the number of uncensored blocks, we find that among the ten block proposers displayed, Bitcoin Suisse and Figment never included deposits and withdrawals to TC's contracts within the analyzed period. Both proposers account for almost 3% of the total number of blocks proposed. We can probabilistically infer that both entities engage in censoring by excluding TC transactions from their blocks. Note, that block proposers adopting PBS with MEV-Boost largely depend on blocks from external block builders.

**Block Relayer Censorship.** Third, we analyze block relayers who intermediate between block builders and block proposers. In Figure 5, we visualize the existing block relayers and the number of blocks forwarded to block proposers that were eventually added to the blockchain. Relayers simulate blocks received from builders, censoring the network by only forwarding blocks that do not include interactions with SDN addresses to block proposers.

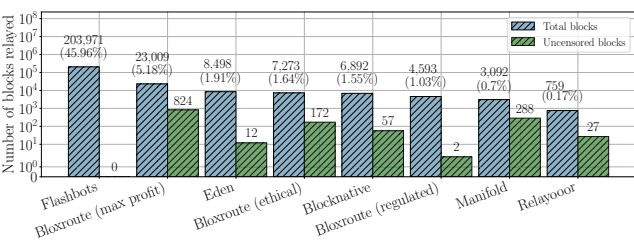

**Figure 5: Ethereum block relayers from PBS activation at block 15,537,940 (Sep-15-2022 08:33:47 AM) to block 15,978,869 (Nov-15-2022 12:59:59 UTC). Uncensored blocks contain interactions with TC contracts.**

At the time of writing, 85% of blocks pass one of the depicted relayers. On Ethereum, there are 8 relay services, three operated by BloXroute. Flashbots relays 79% of blocks. BloXroute's "max profit" relay comes second with a market share of 8.9%. The remaining relayers have a market share between 3.3% and 0.3%. Since PBS activation, Flashbots relayed ~ 46% of proposed blocks. Like their block builders, Flashbots' relay hasn't forwarded blocks with TC transactions, whereas other relayers have.

Concluding, we find that PBS impacts censorship on Ethereum. Block builders and block relayers impose censorship on proposers who are using MEV-Boost. PBS enables block proposers to boost their profits by additionally capturing the MEV in the proposed blocks. As the most successful block builders and block relayers censor TC transactions, block proposers must decide whether to adopt censorship or exclusively connect to a non-censoring relay.

While the censoring block proposers Bitcoin Suisse and Figment both use MEV-Boost, for Bitcoin Suisse we find that only 0.28% of their 9,271 blocks were built by external PBS block builders. Blocks proposed by Bitcoin Suisse were relayed by Blocknative, BloXroute ("max profit"), BloXroute ("regulated"), and Eden. Notably, while BloXroute ("max profit") does not censor TC transactions, there were no TC transactions in the blocks that were relayed by BloXroute ("max profit") and eventually proposed by Bitcoin Suisse. For Figment, 96.8% of the 8,400 blocks were built by third-party block builders from censoring relayers (i.e., Flashbots and BloXroute ("regulated")).

### 5.4 Application Layer Censorship

To quantify censorship at the application level (censorship by smart contract), we focus on a set of smart contracts that include functions to lock or freeze assets (cf. Figure 6). These contracts were deployed to the Ethereum blockchain but are controlled by the entity that deployed them, introducing trust requirements. Figure 6 shows per month the number of newly censored addresses by these contracts. Between January $1^{st}$ and November $15^{th}$, the USDT contract blocked 556 accounts. This exceeds the 87 blocked accounts at USDC. For the stablecoins BUSD and TUSD we find that both have not blocked any address. For USDP, one account has been blocked, labeled as the "Wintermute Exploiter" on Etherscan, a DeFi protocol that was exploited in September 2022 for 160 million USD [18]. For cbETH, we find that a total of 137 accounts have been blocked since its deployment in February 2022. Among the accounts blocked from interacting with the cbETH contract, we identify TC's contracts and

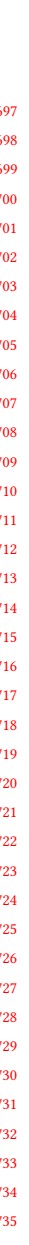

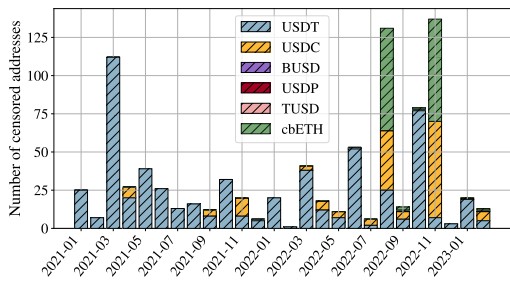

**Figure 6: Censored application layer accounts per month.**

other OFAC-sanctioned entities. In August 2022, with sanctions on TC, blocked addresses hit 131, marking an 84.99% rise from the monthly average between July 2021 and August 2022.

Censorship at the smart contract level can utilize third-party contracts. However, its effectiveness is debatable as sanctioned entities might atomically transfer assets and use alternative accounts.

## 6 SECURITY IMPLICATIONS OF CENSORSHIP

In the following, we explore to what degree censorship affects blockchain security. In line with intuition, censorship is slowing down transaction confirmation latency, which was shown to affect double-spending resilience [29] adversely. Finally, we share an impossibility result on censorship-resilience under PoS.

### 6.1 Transaction Confirmation Latency

**Transaction Latency and Security.** Research indicates that extended mempool presence facilitates double-spending of zero confirmation transactions [29]. Increased confirmation latencies raise the success rate for sandwich attackers targeting trades [83]. Moreover, price shifts in automated market makers can trigger transaction failures if transactions confirm "slowly" [82]. Finally, systematically increased transaction latencies bear the risk of congesting the mempool, increasing the likelihood of transaction re-transmission and P2P network congestion. Congestion slows down block and transaction propagation, deteriorating blockchain security [23, 34].

**Transaction Issuance Time.** We adapt geth [82, 83], Ethereum's predominant client [17], to log all P2P transactions from April to November 2022. A node's observed transactions scale with peer connections, bandwidth, and computing power. Our geth runs on an Ubuntu 20.04.2 LTS, AMD Ryzen Threadripper 3990X (64-core, 2.9GHz), 256 GB RAM, and NVMe SSDs, with a cap of $1,000$ peer connections, up from the standard 50. Located in Europe, it recorded 316.5 million transactions during this period.

**Transaction Confirmation Time.** We rely on the timestamp data recorded in the block header to estimate the transaction confirmation time. It should be noted that this is a rough estimate of the confirmation time because miners and validators (before and after the Ethereum merge, respectively) may decide not to report the precise timestamp when the blocks are generated at. For example, related works have identified evidence of miner misbehavior in block header timestamps for financial gain [78].

**Results.** After gathering the timestamps when a transaction emerges on the P2P network and the time the transaction is included on-chain, we can identify the relative time it takes for a transaction's inclusion. Further, we distinguish between transactions that are and are not subject to censorship (i.e., TC and non-TC transactions). To ensure a fair comparison, we only consider uncensored transactions that are mined in the same blocks as TC transactions at a similar gas price (i.e., ±10%). Two insights emerge:

(1) The time distribution of censored vs. non-censored transactions indicates that censored transactions remain in the mempool longer and confirm slower on-chain. As of November 2022, non-censored transactions average an inclusion delay of 8.7 ± 8.3 seconds, whereas TC transactions average 29.3 ± 23.9 seconds.

(2) The inclusion latency of transactions has grown since Ethereum transitioned to PoS, and the adoption of PBS. For example, the average inclusion delay for TC transactions increased from 15.8 ± 22.8 seconds in August 2022 to 29.3 ± 23.9 seconds in November 2022.

### 6.2 Supplementary Measurement

Blockchain mempool data appears to be heavily location-dependent. Due to network latency, a blockchain node may capture transactions late if they are submitted from a place physically far from the current node's location. For example, European nodes might observe TC transactions faster than others in America. In the worst case, a node may "miss" a transaction if it is included swiftly in the blockchain. Relying on a single node could weaken the conclusions presented in § 6.1. Therefore, we additionally collect data from a supplementary node in the US and compare the results from our main node (in Europe) that we used in § 6.1. We believe that having the mempool data from two different sources across continents corroborates our findings.

We evaluate the effect of a node's geographic location by comparing the transactions observed over a given timeframe. We collected transactions seen by our nodes for 15 days, from February 20th to March 7th, 2023. The supplementary US node uses geth's default settings. Our results remain unchanged when raising the number of peer connections at the US node from 50 to 1,000. Table 1 summarizes the number of blockchain transactions each node observed.

In total, 14,588,990 transactions were included in Ethereum blocks over our data collection period, and both nodes observed more than 95% of these transactions, before their inclusion in a block. Our (main) European node observed some transactions (about 4,000 per day) that the US node did not. On the contrary, only 0.0005% of transactions were only observed by the supplementary US node—and not by the European node. Our results imply that our main node is more powerful than the supplementary node. Also, both nodes failed to observe 3.5% of all transactions. These are likely to be private transactions directly submitted to validators without going through the public mempool [54]. The number of private transactions seems to be about the same as the study by Lyu *et al.* [37], but this ratio is much higher for TC transactions (16.8%).

We also calculate the inclusion time for both nodes (i.e., the difference between the time a node observes a transaction, and the timestamp of the block the transaction was eventually included

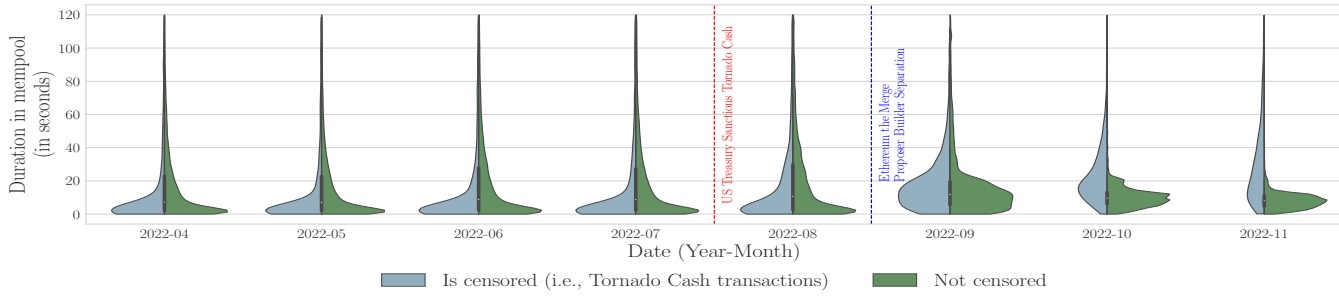

**Figure 7: Transaction inclusion times before and after the merge (15th September 2022), PBS (15th September 2022), and sanctions (8th August 2022). These events notably impacted inclusion delays; e.g., average delay for TC transactions rose from 15.8 ± 22.8 seconds in August 2022 to 29.3 ± 23.9 seconds in November 2022.**

in). More than 99% of the time, the observed inclusion time for the Europe node is lower than that for the US node. The difference is lower than 1 second for more than 99% of observed transactions, and under 0.1 seconds for 30% of them. The difference is statistically significant based on the paired $t$-test (p-value: ≪0.01). The European node and the US node have slightly different ways of recording transactions, which may have affected the time inclusion. The European node records the transaction before it is in the mempool, whereas the US node observes it after, which may have caused the Europe node to observe the transaction first.

We also compare transaction observation time *between* the two nodes and check if there is a statistically significant difference between TC and non-TC transactions. In other words, we test if one node observes TC transactions relatively faster than the other. The result suggests that the difference is statistically insignificant across TC and non-TC transactions, based on the t-test (p-value: 0.116). Furthermore, for both nodes, TC transactions take more time to be included than non-TC transactions with a significant margin (p-value: ≪0.01), thereby corroborating the conclusions of § 6.1. This time, we did not control for confounding factors such as gas fees. For all the statistical tests discussed here, we set the threshold to 5% and excluded outliers (i.e., transactions that take more than 120 seconds to be included in a block). Hence, the observations of our main node are reliable enough for the analyses in § 6.1.

## 7 DISCUSSION

Our analysis indicates that distinguishing individual actors is non-trivial, as the behavior of one can influence others' practices. Block proposers utilizing MEV-Boost rely on block builders and relayers for payload delivery. Consequently, these proposers, who by default use a profit-maximizing strategy to choose payloads, often accept assigned blocks without assessing potential contributions to censorship. By building blocks locally (as was the standard before PBS) or by exclusively connecting to non-censoring relays, block proposers can ensure to not partake in censorship. Furthermore, proposers can use the *min-bid* flag offered by the MEV-Boost software that enables them to automatically fall back to uncensored block building ("vanilla" building) if the payments offered by blocks constructed by builders are not above a certain threshold [27].

For external block builders who want to maximize the number of blocks they create that are added to the blockchain, censorship by

relayers may push them towards producing censored blocks. For example, if the majority of block proposers are exclusively connected to a single relay, builders are forced to comply with the censoring practices of the respective relay to get their block payloads delivered. Similar to the well-known *feather forking* attack, the financial profit that can be gained by creating blocks that conform with the censorship practices of relayers can pressure external block builders and MEV searchers to comply, as well.

Concerning execution delays, we argue that the consequences depend heavily on the individual application. For TC, a delay in the inclusion and execution of a transaction may not significantly impact the user experience or the contract itself because there are no deadlines within the contract that might cause transactions to revert if they are not incorporated into a block promptly. For more time-dependent contracts such as decentralized exchanges, execution delays may significantly impact user experience. For security reasons, many DeFi contracts include functionalities that protect users from executing transactions that ultimately entail considerable disadvantages in execution price because too much time has passed between the desired and the actual execution. So, although the current censorship practices might not significantly impact the end user, it strongly depends on the individual application.

## 8 CONCLUSION

In this paper, we investigate the impact of censorship in blockchains. We present a systematization of blockchain censorship through formal definitions and quantification of the effect of OFAC sanctions on censorship in Ethereum and Bitcoin. After transitioning to PoS, we find that 46% of Ethereum blocks were made by censoring actors intending to comply with OFAC sanctions. Additionally, we reason about their impact on blockchain security. We find that censorship prolongs the time until a transaction is confirmed, which degrades blockchain security. Finally, we prove that a blockchain cannot be censorship resilient if > 50% of validator nodes enforce censorship.

Our results show that censorship on blockchains is not a mere hypothetical threat: it already degrades the security of existing blockchains and the quality of service for users. Our work sheds light on a dilemma anticipated for a decade: will the promise of a permissionless, secure append-only ledger withstand if regulators intervene? We hope that this work draws attention to the significance of censorship in permissionless blockchains and engenders future work on addressing the mentioned security issues.

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

## A CENSORSHIP ON BITCOIN

Several privacy-enhancing technologies exist on Bitcoin, such as centralized Bitcoin mixers and CoinJoin wallets [62, 76]. All were developed to prevent observers from tracing money flows through the ecosystem, enabling users to increase their on-chain privacy. In contrast to Ethereum, where shared addresses are used to obfuscate money flows, the UTXO-based Bitcoin blockchain relies on shared transactions among users. In the following, we exclusively focus on the centralized Bitcoin mixer Blender.io, since CoinJoin Wallets, such as Wasabi Wallet or Samurai Wallet, were not targeted by

**Table 1: Number of blockchain transactions observed by each our Ethereum nodes.**

| | US observed | US not observed |
|---|---|---|
| **Europe observed** | 14,024,697 (96.1%) | 58,742 (0.4%) |
| **Europe not observed** | 86 (0.0%) | 505,465 (3.5%) |

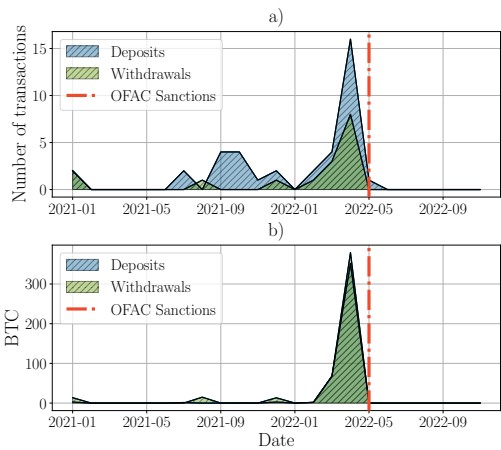

**Figure 8: Interactions with the Bitcoin mixer Blender.io.**

OFAC sanctions. Blender.io was sanctioned by OFAC in May 2022, as discussed above.

In Figure 8, we visualize the interactions with the sanctioned Blender.io over time. Figure 8 a) shows the number of transactions with Blender.io from January 2021 to November 15th, 2022. In Figure 8 b), we display the amount of BTC deposited and withdrawn for the same period. We observe that after OFAC's sanctions against Blender.io were imposed, there were no further interactions with the application. Shortly before the sanctioning, there was an observable spike in deposits and withdrawals from Blender.io. We find that 351 BTC and 379.06 BTC were deposited and withdrawn from addresses belonging to Blender.io the month before the sanctioning. Assuming an exchange rate of 35000 USD per BTC, around $10.5 million were deposited and withdrawn in a single month, just before the sanctions took effect.

Figure 8 suggests that the sanctions entirely prevented Blender.io from continuing to provide its centralized services. We do not propose that this occurred due to censorship as defined in Definitions 2-4, as the likely cause is the removal of the Blender.io website.

## B IMPOSSIBILITY OF CENSORSHIP-RESILIENCE

In this section, we argue that previous results on liveness in PoS [20, 57, 60] constitute a lower bound for censorship-resilience on the consensus layer. Concretely, we prove that no PoS protocol can achieve *censorship resilience* if the number of censoring validators makes up more than 50% of the validator committee. In the following, we first introduce our model in reasoning about security in PoS blockchains. We outline recent results on *liveness* in PoS

blockchains and further introduce the relationship of censorship-resilience to liveness. After introducing an intuition, we state our impossibility result in Theorem 7 and prove it through a sequence of worlds and an indistinguishability argument.

**Security Model.** Recall that $B_i \in \mathcal{L}$, if a block $B_i$ is included in the distributed ledger $\mathcal{L}$, and that $tx$ is included in $\mathcal{L}$ if $tx \in B_i \wedge B_i \in \mathcal{L}$ (cf. Section 4.2). Two views of ledgers $\mathcal{L}_1, \mathcal{L}_2$ are conflicting if they differ in their included transactions. We further assume that $n$ is the total number of validators. In our model, transactions $tx$ are input to validators by the environment $\mathcal{Z}$. Before the execution of the protocol starts, an adversary $\mathcal{A}$ corrupts a subset of validators $f < n$ and renders them *adversarial* such that they can arbitrarily deviate from the specified protocol. The remaining validators are *honest* and follow the protocol as specified. We assume that network communication is synchronous, hence messages are instantly delivered once they are sent by a network participant.

**Safety & Liveness.** The safety and liveness of Proof-of-Stake blockchains were studied under varying synchronicity assumptions [20, 22, 60], and assuming dynamic validator committees [47]. However, definitions of liveness subtly differ in their phrasing. Whereas some works focus on the eventual inclusion of a transaction in a block after a finality delay $\Delta$ [41, 47, 60], others focus on the correct report of a value upon a query sent by an honest client [22]. To formally define liveness, we follow the holistic definition of Garay *et al.* [21], which states that all transactions originating from an honest client will eventually end up in an honest validators' view of a ledger, hence an adversary cannot perform a DoS attack against honest clients. We formally define properties of liveness in a PoS protocol as follows.

*Definition 5.* A validator ensures *liveness* of a PoS protocol if it satisfies the following properties:

(1) **Propagation.** Upon reception of a transaction $tx$ by an honest client $C$, the validator forwards $tx$ to other peers in the network.

(2) **Inclusion.** A transaction $tx$ sent by an honest client $C$ is eventually included in the local view of an honest validator's distributed ledger $\mathcal{L}$.

(3) **Availability.** Upon query, an honest validator will report whether a transaction is included in the ledger.

Importantly, recent results highlight a trade-off between accountability and availability [47] and show the impossibility of liveness beyond $f > \frac{n}{2}$ adversarial validators [60].

**Modeling Censorship.** In a real-world environment, a validator censoring transactions may not be considered adversarial. For example, censorship may be considered beneficial from a legal perspective, as malicious actors are prevented from participating in the system. However, as first identified by Miller *et al.* [41], censorship-resilience is a property of liveness. We further argue that the act of censorship is equivalent to a subset of adversarial actions as defined in a PoS protocol. To formally define this finding, we say that $\mathcal{A}_C$ is a probabilistic polynomial time algorithm, where a subset $f_c < n$ of validators, which are corrupted by $\mathcal{A}_C$, arbitrarily deviate from the PoS protocol. For example, a censoring validator may, upon reception of a transaction $tx$ by the environment, $\mathcal{Z}$ by

**Table 2: TC contracts we capture in this work.**

| TC Contract | Address |
|---|---|
| 0.1 ETH | 0x12D66f87A04A9E220743712cE6d9bB1B5616B8Fc |
| 1 ETH | 0x47CE0C6eD5B0Ce3d3A51fdb1C52DC66a7c3c2936 |
| 10 ETH | 0x910Cbd523D972eb0a6f4cAe4618aD62622b39DbF |
| 100 ETH | 0xA160cdAB225685dA1d56aa342Ad8841c3b53f291 |
| 0.1k DAI | 0xD4B88Df4D29F5CedD6857912842cff3b20C8Cfa3 |
| 1k DAI | 0xFD8610d20aA15b7B2E3Be39B396a1bC3516c7144 |
| 10k DAI | 0x07687e702b410Fa43f4cB4Af7FA097918ffD2730 |
| 100k DAI | 0x23773E65ed146A459791799d01336DB287f25334 |
| 5k CDAI | 0x22aaA7720ddd5388A3c0A3333430953C68f1849b |
| 50k CDAI | 0x03893a7c7463AE47D46bc7f091665f1893656003 |
| 500k CDAI | 0x2717c5e28cf931547B621a5dddb772Ab6A35B701 |
| 5m CDAI | 0xD21be7248e0197Ee08E0c20D4a96DEBdaC3D20Af |
| 100 USDC | 0xd96f2B1c14Db8458374d9Aca76E26c3D18364307 |
| 1k USDC | 0x4736dCf1b7A3d580672CcE6E7c65cd5cc9cFBa9D |
| 10k USDC | 0xD691F27f38B395864Ea86CfC7253969B409c362d |
| 100 USDT | 0x169AD27A470D064DEDE56a2D3ff727986b15D52B |
| 1000 USDT | 0x0836222F2B2B24A3F36f98668Ed8F0B38D1a872f |
| 10k USDT | 0xF67721A2D8F736E75a49FdD7FAd2e31D8676542a |
| 100k USDT | 0x9AD122c22B14202B4490eDAf288FDb3C7cb3ff5E |
| 0.1 WBTC | 0x178169B423a011fff22B9e3F3abeA13414dDD0F1 |
| 1 WBTC | 0x610B717796ad172B316836AC95a2ffad065CeaB4 |
| 10 WBTC | 0xbB93e510BbCD0B7beb5A853875f9eC60275CF498 |

refusing to *(i)* include *tx* in block $B_i$, *(ii)* propagate *tx* to other peers in the network *(iii)* build upon $\mathcal{L}$, where *tx* is included in $\mathcal{L}$, and by *(iv)* refusing to attest to $B_i$, where $tx \in B_i$. Given the previous reasoning, we define Δ-*censorship-resilience* as follows.

*Definition 6.* (Δ-Censorship-resilience) Suppose an honest client inputs *tx* to $(n - f_c)$ honest validators. Then, *tx* is committed to $\mathcal{L}$ within Δ, except with negligible probability.

**Intuition.** Let us consider a PoS protocol where $f_c > \frac{n}{2}$ of validator nodes are directly censoring transactions. We show that censoring impacts the liveness of a blockchain. Intuitively, censorship of a transaction prevents it from being included in the blockchain, as a censoring validator drops transactions that, e.g., involve sanctioned addresses. As such, censoring validators create a conflicting chain, which can be considered *adversarial* in the context of traditional Byzantine Fault Tolerant consensus protocols. To prove Theorem 1, we show that the threat to liveness posed

by a validator corrupted by $\mathcal{A}$ is indistinguishable from the threat to liveness posed by a validator corrupted by $\mathcal{A}_C$, hence that is censoring. We defer the proof of liveness to the argument presented by Tas *et al.* [60] and present Theorem 7.

THEOREM 7. *Consider a PoS protocol* Π *with n validators in a synchronous network, where at least $f_c > \frac{n}{2}$ are corrupted by $\mathcal{A}_C$. Then,* Π *cannot provide censorship-resilience.*

**Proof.** Suppose the number of validators is $n \in \mathcal{Z}$, where we assume that *n* is even in each epoch. Further, consider there exists a protocol Π that supports *liveness* with $f < \frac{n}{2}$ corrupted validators that is further Δ-*censorship-resilient* with $f_c > \frac{n}{2} - f$ censoring validators. Then, there exists a decision function $\mathcal{D}$, which outputs a non-empty set of censoring validators. We prove Theorem 7 through a sequence of worlds and an indistinguishability argument. *(World 1.)* Let $P$, $Q$ and $R$ partition *n* into three disjoint groups, where $|P| < \frac{n}{2}, |Q| > \frac{n}{2} - |P|$ and $R = n - |P| - |Q|$. Nodes in $P$ are corrupted by $\mathcal{A}$, nodes in $Q$ are corrupted by $\mathcal{A}_C$ and nodes in $R$ are honest. Corrupted nodes are adversarial and do not communicate with honest nodes in $R$. Hence, upon input of randomly distributed transactions by $\mathcal{Z}$, validators in $P \cup Q$ output a diverging view of $\mathcal{L}$ as opposed to $R$. So the decision function outputs a non-empty set of adversarial validators $P \cup Q$.
*(World 2.)* Let $P$, $Q$ and $R$ partition *n* into three disjoint groups, where $|P| < \frac{n}{2}, |Q| > \frac{n}{2} - |P|$ and $R = n - |P| - |Q|$. Nodes in $P$ are corrupted by $\mathcal{A}_C$, nodes in $Q$ are corrupted by $\mathcal{A}$ and nodes in $R$ are honest. Corrupted nodes are adversarial and do not communicate with honest nodes in $R$. Hence, upon input of randomly distributed transactions by $\mathcal{Z}$, validators in $P \cup Q$ output a diverging view of $\mathcal{L}$ as opposed to $R$. So the decision function outputs a non-empty set of adversarial validators $P \cup Q$.

However, worlds 1 and 2 are indistinguishable for the decision function $\mathcal{D}$. Thus, $\mathcal{D}$ cannot output a non-empty set of censoring validators. □

We note that this lower bound for censorship-resilience also applies to Nakamoto consensus [21, 43].

