# OpenReview forum: "Blockchain Censorship"
_ACM.org/TheWebConf/2024/Conference — TheWebConf24_

### Official Review · Reviewer_WgVY · 2023-11-09

**Novelty:** 6
**Technical Quality:** 5

**Review:**

The paper considers the effect of *censorship* on permissionless blockchains, where restrictions are placed on how entities involved in running the blockchain interact, for example as a result of government sanctions.
The bulk of the paper is an empirical study of the effect of recent sanctions on the Ethereum blockchain, which is of interest since permissionsless blockchains are designed to be resistant to censorship, showing impact on service quality and a potential threat to security.
In addition there is a theoretical contribution consisting of formalizing notions of blockchain censorship and a negative result showing (not surprisingly) that censorship resistance is impossible if the majority of validator nodes are affected by the censorship.

I found that the paper raises an important question and provides a foundation on which further work could be based. The theoretical contribution is not very deep, but the empirical study is interesting.

**Questions:**

Your conclusion suggests that more work is needed to address threats due to censorship. Can you discuss to what extent the impact of censorship you observed may be *unavoidable* for any protocol or a property of the specific protocol studied?

Can you discuss the relationship between your model and models of adversarial attacks studied in neighboring peer communities (e.g. https://www.sciencedirect.com/science/article/abs/pii/S1367578822000049)? This is not my area of expertise, but the discussion of related work seemed a bit narrow.

**Reviewer Confidence:**

2: The reviewer is willing to defend the evaluation, but it is likely that the reviewer did not understand parts of the paper

**Scope:**

3: The work is somewhat relevant to the Web and to the track, and is of narrow interest to a sub-community

---

### Official Review · Reviewer_2GMr · 2023-11-17

**Novelty:** 5
**Technical Quality:** 3

**Review:**

Pros:
- The paper studies an important problem on how censorship affects the ecosystem of decentralized blockchains.
-The study shows that censorship slows down the confirmation latency of block building, which is a strong evidence.
- The paper is well-written.

Cons:
- My major concern is on the main takeaway of this paper. Empirically, this work shows in Section 5.2 that the censorship largely affects TC transactions, and presents in Section 6 that the inclusion of censorship increases the confirmation latency for both censored and non-censored blocks. These results are interesting. However, the studies in Section 5.2 and 5.3 are a bit unclear to me. I am not sure that the implication on the data for specific builders/proposers/relayers/applications. It would be better to me if the authors could present typical flows of different blocks with different builders/proposers/relayers and investigate how censorship affect (the fraction of) these flows. Meanwhile, the definitions in Sections 4.2 and 4.3 seem to have little relationship with the major parts of the main body. The same for Appendix B.
- Detailed issues.
	- In the Introduction, the authors can discuss more about the responsibilities of block builders/proposers/relayers.
	- Wrong citation places in Lines 204, 207, 210, 219, citation should be placed before the period.
	- In Definitions, what does the notation "|" mean? Further, do we need the subscript $i$ here?
	- For the date expression, in Line 519, the authors use July 1st; in Line 571, they use November 15; in Line 600, they use $15^{th}$. These should be unified.
	- In Line 517, the authors use "ten largest block builders"; there are twelve of them in Figure 3.
	- In Line 670, "9,271" should be "9,274"?

**Questions:**

See the above cons. Also, I have the following further questions.
1. What is the difference between strict and weak censorship? Can you elaborate more?
2. In Figure 1, 3rd subfigure, what does "TORN" mean here for the y-label? Further, the dashed-dotted line space in this subfigure stands for a different date compared with the other two. Can you explain?
3. Can you explain why Ethermine is affected more significantly by censoring, as in Figure 2?
4. In the theoretical part of Appendix B, you say that "censorship-resilience is a property of liveness." Are you regarding censorship as a type of adversarial behavior towards the blockchain? I am expressing nothing political here.

**Reviewer Confidence:**

3: The reviewer is confident but not certain that the evaluation is correct

**Scope:**

4: The work is relevant to the Web and to the track, and is of broad interest to the community

---

### Official Review · Reviewer_m4Sb · 2023-11-24

**Novelty:** 6
**Technical Quality:** 6

**Review:**

The paper assesses the impact of US sanctions on two widely used cryptocurrency mixing services, namely Blender.io and Tornado Cash, with a specific focus on the consensus layer. Through the collection of relevant blockchain transactions, the authors formalise and quantify the impact on various aspects, including the transaction volume, the miners, block builders, proposers, relayers, etc. They also analyse the censorship implications on blockchain security.

I enjoyed reading the paper and found many interesting insights. The paper is well-written, easy to follow, and has several strengths. It offers timely and significant insights into the intervention impacts. The chosen methods and datasets are appropriate. Data are collected from Bitcoin and Ethereum blockchains, thus are likely to be complete. The literature review is good, covering a wide range of other censorships such as eclipsing the Bitcoin and Ethereum P2P networks.

**Questions:**

I only have a few questions that might help improve the paper.

1. Figure 1 does a good job of describing the number of deposits and withdrawals, but how does it look like in terms of transaction value?
2. Given the substantial size of the quantitative dataset, wouldn't it be beneficial to conduct statistical tests in Section 5 to enhance the robustness of the findings? The authors employed this approach in Section 6.
3. Is there any discernible impact of censorship on Tornado Cash Nova, the successor to Tornado Cash that allows arbitrary deposits?
4. It would be great if they extend the timeframe beyond 2022 to capture more recent data.
5. While blockchains are designed for decentralisation, it appears that they can become highly centralised around a limited number of nodes and validators. Can you discuss more about that?

**Reviewer Confidence:**

3: The reviewer is confident but not certain that the evaluation is correct

**Scope:**

4: The work is relevant to the Web and to the track, and is of broad interest to the community

---

### Official Review · Reviewer_voa5 · 2023-11-28

**Novelty:** 3
**Technical Quality:** 4

**Review:**

The authors study the effect of censorship on permissionless blockchains. For this, they define what censorship means at different layers such as network, consensus and application layer. They then provide an empirical quantification of censorship on Ethereum and Bitcoin.

I would appreciate if the authors provided clarification regarding the objective of the paper. The authors seem to provide extensive statistical data about how censorship has affected blockchain during various times. For this, they first provide definitions of censorship at various layers. However, the lack of sufficient background on blockchain operations and the components involved in block inclusion makes it challenging to understand the scope of these definitions. For instance, it is not clear what a block height means. Further, the paper uses a lot of blockchain jargon where most of the terms are not defined in the paper. While there exist definitions for a few of these in the ‘notations and terminology’ section in prelims, the placement of this section at a much later point in the paper adds to confusion until this section is reached. Further, it is not clear what the authors are trying to capture when they say, ‘empirical quantification’. The section on ‘censorship quantification’ which provides this quantification is essentially giving statistical data about how the inclusion of censorship has affected the time taken to include a block in the blockchain. Next, based on the title of the section on ‘security implications of censorship’, I got the impression that it will contain some discussion about how privacy issues have arisen due to the inclusion of censorship. On the contrary, this section too talks about how the transaction latency is affected due to censorship. I do not see any discussion about how blockchain security is affected, and hence, I believe that this section heading is misleading. Further, the authors also prove that a blockchain cannot be censorship resilient if more than half of the validator nodes enforce censorship. Although this is claimed as a contribution, it is provided as part of the supplementary material in the appendix. I would appreciate clarification on the reasoning for moving this proof to the appendix and how it affects the results presented in the paper.

I would appreciate additional clarity regarding the authors' objectives. There is a lot of statistical data that is provided without providing clear inferences. Even in places where conclusions are drawn, it is not clear why these are significant. Further, the provided definitions of censorship lack clarity, particularly without the necessary background information on blockchains.

**Questions:**

- I suggest the authors add more background in the prelims section to facilitate a novice reader in understanding the necessary details about blockchains. Further, it would be beneficial to provide information about any blockchain-related terms that appear in the text before the prelims section. This would enhance the reader's understanding and prevent confusion due to unfamiliar terminology.

- The authors should clarify what they are trying to capture when they mean ‘censorship quantification’. Further, it will be helpful if the authors can clearly demarcate the observations they make based on all the statistical data that is reported, from the conclusions/inference that they draw. This will enhance the readability. I would also suggest that the authors make it clearer regarding the goal they are trying to achieve in the process of this ‘quantification’.

- I would appreciate further clarification on the security implications the authors aim to address in Section 6. The section seems to present statistical data with observations and inferences, and I suggest considering a more appropriate title for this section to accurately reflect its content.

- The proof about blockchain not being censorship resilient when majority of the validator nodes enforce censorship is present in the appendix. If they authors believe this is a significant contribution, I suggest they bring this in the main body of the paper and swap something else in its place in the appendix. I did not go over this proof since it was not a part of the main body.

**Reviewer Confidence:**

2: The reviewer is willing to defend the evaluation, but it is likely that the reviewer did not understand parts of the paper

**Scope:**

3: The work is somewhat relevant to the Web and to the track, and is of narrow interest to a sub-community

---

### Official Review · Reviewer_dDgn · 2023-11-30

**Novelty:** 4
**Technical Quality:** 3

**Review:**

In the proposed work, authors claim to formalize, quantify and analyze the security impact of blockchain censorship. The topic of this is novel and may be significant in blockchain permissionless systems, there is a significant problem on the clarity of the problem.

Strengths:
New and interesting topic

Weaknesses:
The introduction lacks clear problem definition and motivation, with terms like "blockchain censorship" and "censorship resilience" being frequently used before their formal definition in Section 3, resulting in confusion early on.
Additionally, there is a delayed explanation of TC's motivation, which is not addressed until the background section where TC is heavily used in the introduction and throughout till quantification section.
The rationale behind using Blender.io also remains unclear.
The introductory section is cluttered with a series of events that lack proper citations and context. Moreover, the absence of illustrative figures for the system model and the lack of foundational information on the work of Zhou et al., which underpins the model's extension, further detract from the paper's clarity.

**Questions:**

I would like authors to address weakness given above.

**Reviewer Confidence:**

4: The reviewer is certain that the evaluation is correct and very familiar with the relevant literature

**Scope:**

3: The work is somewhat relevant to the Web and to the track, and is of narrow interest to a sub-community

---

### Official Review · Reviewer_iQD1 · 2023-12-02

**Novelty:** 5
**Technical Quality:** 6

**Review:**

This paper investigates the impact of censorship on permissionless blockchains, focusing on the security implications. The study begins by defining blockchain censorship and then quantitatively assesses current practices, finding that 46% of Ethereum blocks were created by censoring actors in compliance with OFAC sanctions, highlighting the significant influence of these sanctions on blockchain neutrality. The paper also explores how censorship affects security, particularly after Ethereum's transition to Proof-of-Stake (PoS), where censored transactions experienced an average delay of 85%, increasing their vulnerability and enhancing risks from sandwich attacks. The authors demonstrate that censorship not only compromises transaction neutrality but also blockchain security, with censored transactions facing extended delays in the memory pool. The paper concludes by proving a fundamental limitation in PoS and Proof-of-Work (PoW) protocols, asserting that no PoS or PoW blockchain can achieve censorship resilience if more than 50% of validator nodes or miners engage in direct censorship.

This study provides a comprehensive analysis of the implications of applied censorship measures on blockchain systems. A couple of issues I have with the paper as it currently stands are the following:
- The paper states that they are expanding the model introduced by [84] without clearly describing their contribution and the shortcomings in [84] that requires this expansion.
- The dataset and samples used are pretty old. I'm not sure if more recent examples of censorship exists but it would have been nice to see more recent examples of such censorship and its effects.

**Questions:**

- Explain how you have expanded the model presented in [84]
- What would be the reason for a node to do weak censorship instead of strong censorship?
- Are there more recent examples of censorship enforced by OFAC?

**Reviewer Confidence:**

2: The reviewer is willing to defend the evaluation, but it is likely that the reviewer did not understand parts of the paper

**Scope:**

4: The work is relevant to the Web and to the track, and is of broad interest to the community

---

### Decision · Program_Chairs · 2024-01-22

**Decision:**

Accept

**Comment:**

Summary: Study that formalizes, quantifies, and analyzes the security impact of blockchain censorship.


 Strengths:
 + Impact of censorship on blockchains is relevant and important.
 + Clear analysis
 + Appropriate datasets/methods
 + Some valuable insights into censorship impact
 + Empirical study is interesting and lays a foundation for future research.


 Weaknesses:
 - Concerns about unclear definition/motivation
 - Need better/enough context/relevant citations
 - Some places need better writing & exposition, eg, rationale, figures, objectives
 - Definitions of censorship could be improved
 - Theory not the most novel


 Recommendation: Interesting study but would benefit from better writing. Please fulfill the promises made to the reviewers in the author-response.